# The Role of High-Sensitivity Troponin I in Predicting Atrial High-Rate Episodes (AHREs) in Patients with Permanent Pacemakers

**DOI:** 10.3390/life15121850

**Published:** 2025-12-02

**Authors:** Linh Ha Khanh Duong, Nien Vinh Lam, Vinh Thanh Tran

**Affiliations:** 1Cho Ray Hospital, Ho Chi Minh City 700000, Vietnam; khanhlinh175@gmail.com (L.H.K.D.); thanhvinhtran2002@gmail.com (V.T.T.); 2Department of Biochemistry, Faculty of Medicine, University of Medicine and Pharmacy at Ho Chi Minh City, Ho Chi Minh City 700000, Vietnam

**Keywords:** high-sensitivity troponin I, atrial high-rate episodes, ahre, permanent pacemaker, biomarkers, NT-proBNP, subclinical atrial fibrillation

## Abstract

**Background:** Atrial high-rate episodes (AHREs) detected by pacemakers are linked to increased stroke risk. The predictive value of high-sensitivity cardiac troponin I (hs-cTnI) for AHREs in pacemaker patients remains uncertain. This study evaluated baseline hs-cTnI as a predictor for new-onset AHREs in this population. **Methods:** This prospective cohort study enrolled 272 patients undergoing permanent pacemaker implantation. We excluded 40 patients with pre-existing atrial fibrillation (AF), leaving a total of 232 patients (mean age 63.7 years; 53.4% male) in the at-risk cohort. Baseline hs-cTnI and NT-proBNP were measured. The primary endpoint was new-onset AHREs (>175 bpm), detected by device interrogation over a median follow-up of 12 months. **Results:** New-onset AHREs occurred in 65 (28.0%) patients. Contrary to our hypothesis, baseline hs-cTnI levels did not differ significantly between patients who developed AHREs and those who did not (median 16.5 vs. 15.7 pg/mL, *p* = 0.148). Multivariable Cox regression confirmed that neither hs-cTnI nor NT-proBNP were independent predictors. Instead, Sick Sinus Syndrome (HR 2.10, *p* < 0.001), heart failure (HR 1.78, *p* = 0.010), and Left Atrial Diameter (HR 1.15, *p* = 0.006) were significant independent predictors. **Conclusions:** In this high-risk pacemaker cohort, baseline hs-cTnI and NT-proBNP did not predict short-term new-onset AHREs. Established electrical and structural substrates appear to be the overwhelming drivers of arrhythmia in this specific population.

## 1. Introduction

Atrial fibrillation (AF) is the most common cardiac arrhythmia globally [1,2]. Its prevalence is rising, and it is associated with severe complications, including a significantly increased risk of stroke and heart failure and higher overall mortality [1,3,4]. A significant challenge in managing AF is the existence of subclinical or asymptomatic episodes, often missed by traditional diagnostic methods like standard ECGs, which still contribute to the overall disease burden and risk [5,6]. Patients with cardiac implantable electronic devices (CIEDs), such as permanent pacemakers, offer a unique opportunity for study. These devices can continuously monitor the atrial rhythm, allowing for the detection of asymptomatic atrial high-rate episodes (AHREs), which are considered a form of subclinical atrial fibrillation [5,7] and have been linked to a 2.5-fold increased risk of stroke [8].

Identifying which patients will develop these arrhythmias is crucial for prevention. While traditional cardiovascular risk factors are known, they do not fully account for an individual’s risk of developing AF. This has led to a search for blood-based biomarkers that could improve risk prediction and enhance understanding of the underlying pathophysiology. Various markers reflecting processes like atrial strain, inflammation, and myocardial fibrosis have been investigated [9].

Among the most studied biomarkers, N-terminal pro-B-type natriuretic peptide (NT-proBNP), an indicator of myocardial stretch, has been consistently suggested as a strong predictor for incident AF [10]. Another key biomarker is cardiac troponin, a protein released following myocardial cell injury. While critical for diagnosing acute coronary syndromes, elevated troponin levels are also found in other conditions, including supraventricular tachyarrhythmias. The conditions that lead to the development of AF may also increase troponin levels. Although some community-based studies have confirmed an association between elevated troponin and the subsequent development of AF [11], the overall scientific evidence remains limited. Furthermore, its specific value as an independent risk marker, particularly when compared to established biomarkers like NT-proBNP, is not yet certain.

This study was conducted to explore the role of high-sensitivity troponin I (hs-cTnI) in predicting new-onset atrial high-rate episodes (AHREs) in patients with permanent pacemakers. Subclinical arrhythmias can be accurately detected in this specific population, providing a valuable opportunity to determine whether baseline hs-cTnI levels can serve as a predictor for the future development of AHREs.

## 2. Materials and Methods

### 2.1. Study Design and Population

We conducted a prospective cohort study at the Department of Arrhythmia Treatment and the Department of Biochemistry at Cho Ray Hospital, Vietnam. Adult patients (≥18 years) who had a clinical indication for and agreed to undergo permanent pacemaker implantation between September 2023 and August 2026 were enrolled. All participants provided written informed consent. We excluded patients who were critically ill with a high risk of mortality, pregnant, had an indication for surgery, or had severe renal impairment defined as an estimated glomerular filtration rate (eGFR) of ≤30 mL/min/1.73 m^2^.

A total of 285 patients were screened. Thirteen patients were excluded due to severe renal impairment (eGFR ≤ 30 mL/min/1.73 m^2^). Of the remaining 272 eligible patients, 40 had documented pre-existing atrial fibrillation and were excluded from the prospective analysis. The final analytic cohort consisted of 232 patients free of AF at baseline.

### 2.2. Data Collection and Baseline Assessment

Baseline clinical data, including demographics (age and sex), anthropometrics (body mass index, BMI), and medical history (cardiovascular risk factors such as hypertension, diabetes mellitus, heart failure (HF), and prior stroke), was collected from patient medical records and clinical examinations prior to pacemaker implantation. Standard 12-lead ECGs and transthoracic echocardiogram results, including left atrial (LA) diameter and left ventricular ejection fraction (EF), were also recorded using a Vivid E95 ultrasound system (GE Healthcare, Horten, Norway).

### 2.3. Biomarker Measurement

Blood samples for measuring high-sensitivity troponin I (hs-cTnI) and N-terminal pro-B-type natriuretic peptide (NT-proBNP) were collected from all patients before pacemaker implantation. All assays were performed at the Department of Biochemistry at Cho Ray Hospital using a Siemens ADVIA Centaur XPT automated immunoassay system (Siemens Healthcare Diagnostics, Tarrytown, NY, USA). hs-cTnI was measured using a 3-site sandwich immunoassay, while NT-proBNP was measured using a 2-site sandwich immunoassay, both with direct chemiluminescence technology.

### 2.4. Outcome Definition and Follow-Up

Patients were divided at baseline into those with pre-existing AF (*n* = 40) and those without (*n* = 232). The non-AF group (*n* = 232), which constituted the at-risk population, was followed prospectively for a median of 12 months to monitor for the primary endpoint. The primary endpoint was the first detection of new-onset atrial high-rate episodes (AHRE) or clinical AF. Follow-up was conducted during routine pacemaker interrogation appointments. The diagnosis of an event was based on data from the cardiac implantable electronic device (CIED) or a standard ECG. Clinical atrial fibrillation (AF) was defined according to the 2020 European Society of Cardiology (ESC) guidelines as a 12-lead ECG or a single-lead ECG strip of ≥30 s showing a heart rhythm with no discernible repeating P waves and irregular RR intervals [12]. AHREs were defined as device-detected atrial tachyarrhythmias with a rate > 175 bpm [13]. To ensure diagnostic accuracy, all stored intracardiac electrograms (EGMs) were reviewed by two independent cardiologists to rule out artifacts or far-field R-wave sensing. Disagreements were resolved by consensus or consultation with a third senior electrophysiologist.

### 2.5. Statistical Analysis

All statistical analyses were performed using R software version 4.5.1 (R Foundation for Statistical Computing, Vienna, Austria). Continuous variables were assessed for normality using the Shapiro–Wilk test. Normally distributed data were presented as the mean ± standard deviation (SD) and compared using Student’s *t*-test, while non-normally distributed data were presented as the median and interquartile range (IQR) and compared using the Wilcoxon rank-sum test. Categorical variables were reported as frequencies (*n*) and percentages (%) and compared using the Chi-square test or Fisher’s exact test. The cumulative incidence of new-onset AHREs was analyzed using the Kaplan–Meier method, and differences between groups (based on baseline biomarker quartiles) were assessed using the log-rank test. Multivariate Cox proportional hazards regression was used to identify independent predictors for the incidence of AHREs, including baseline hs-cTnI levels and other clinical, echocardiographic, and laboratory factors. A two-sided *p*-value of <0.05 was considered statistically significant.

### 2.6. Ethical Considerations

The study protocol was approved by the Institutional Review Boards of the University of Medicine and Pharmacy at Ho Chi Minh City and Cho Ray Hospital. The research was conducted in accordance with the Declaration of Helsinki, and participants provided written informed consent prior to any study-related procedures.

## 3. Results

### 3.1. Patient Population and Baseline Characteristics

A total of 285 patients were initially screened for the study. After excluding 13 patients who had an estimated glomerular filtration rate (eGFR) below 30 mL/min/1.73 m^2^, the final study cohort consisted of 272 patients. This cohort was stratified at baseline into 40 patients with pre-existing atrial fibrillation and 232 patients without atrial fibrillation. The latter constituted the at-risk population for analyzing the prediction of new-onset arrhythmias.

The baseline characteristics of this at-risk cohort (*n* = 232) are presented in Table 1. The mean age was 63.7 ± 14.0 years, 124 (53.4%) were male, and the mean body mass index (BMI) was 22.5 ± 3.34 kg/m^2^. Comorbidities were common, with 15.6% of patients having a history of coronary artery disease (CAD). Regarding baseline medication, 40.7% were on statins, 25.1% were on ACE inhibitors or ARBs, and 7.8% were on beta-blockers (Table 1).

For the total cohort (*n* = 232), the primary indications for pacemaker implantation were AV block (34.9%) and Sick Sinus Syndrome (33.1%). The majority of patients (66.8%) received a dual-chamber pacemaker.

### 3.2. Incidence of Atrial High-Rate Episodes (AHREs)

Over a median follow-up period of 12.0 months (interquartile range [IQR] 6.75–13.0 months), 65 of the 232 at-risk patients (28.0%) developed new-onset AHREs as detected by their implanted devices.

The Kaplan–Meier analysis showed a cumulative probability of developing AHREs of approximately 30% at 12 months, which increased to approximately 35% by 17 months (Figure 1). Among the 65 patients who developed AHREs, the duration of the longest recorded episode was less than 30 s in 46% of cases, between 30 s and 6 min in 40% of cases, and greater than 6 min in 14% of cases.

**Figure 1 life-15-01850-f001:**
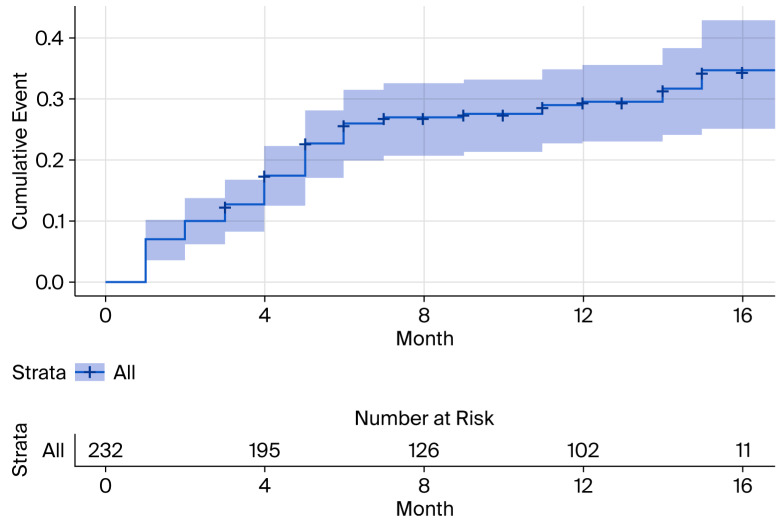
Kaplan–Meier curve showing the cumulative probability of AHREs over time.

### 3.3. Baseline hs-cTnI and NT-proBNP Levels and Risk of New-Onset AHREs

We compared the baseline biomarker levels between the patients who developed and did not develop AHREs during the follow-up period.

The median baseline hs-cTnI level was not statistically different between the two groups. The AHRE group (*n* = 65) had a median hs-cTnI of 16.5 pg/mL (IQR 5.61–80.9) compared to 15.7 pg/mL (IQR 4.06–95.3) in the no-AHRE group (*n* = 167) (Wilcoxon rank-sum test, *p* = 0.148). A Kaplan–Meier survival analysis, stratified by baseline hs-cTnI quartiles, confirmed this finding, showing no significant difference in the cumulative incidence of AHREs across the quartiles (Log-rank test, *p* = 0.9) (Figure 2).

**Figure 2 life-15-01850-f002:**
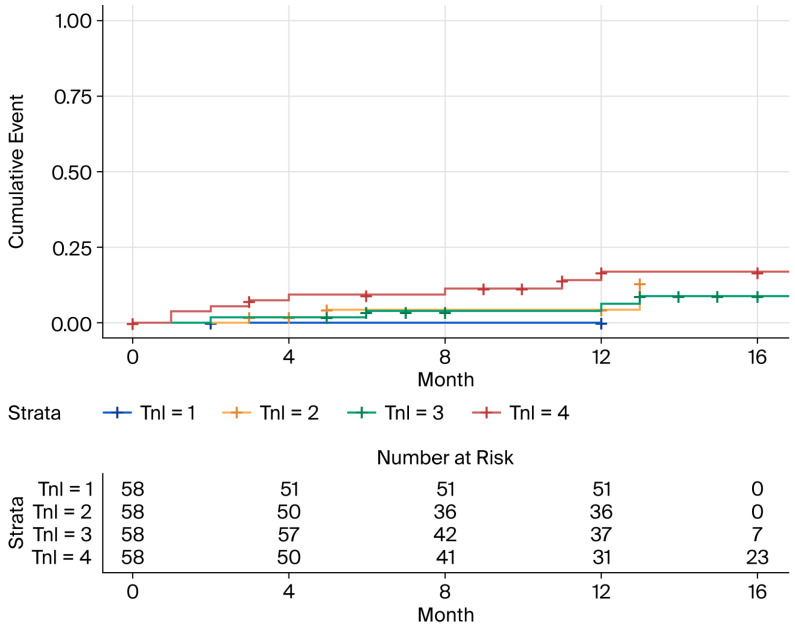
Kaplan–Meier comparison of new-onset AHRE cases by baseline hs-cTnI concentration quartiles.

Similarly, baseline NT-proBNP levels did not predict the development of AHREs. The median NT-proBNP level was 37.1 pmol/L (IQR 10.7–153.7) in the group that developed AHREs compared to 32.9 pmol/L (IQR 7.07–157) in the no-AHRE group (Wilcoxon rank-sum test, *p* = 0.396). The Kaplan–Meier analysis stratified by NT-proBNP quartiles also showed no significant difference in the cumulative event rate (Log-rank test, *p* = 0.2).

### 3.4. Multivariable Cox Proportional Hazards Analysis for Predictors of New-Onset AHREs

We performed a multivariable Cox regression analysis incorporating clinical factors, medications, and biomarkers (Table 2). Even after adjusting for potential confounders, such as coronary artery disease (HR 1.24, *p* = 0.310) and medication use (ACE inhibitors/ARBs and Beta-blockers), baseline hs-cTnI and NT-proBNP levels remained non-significant predictors of AHREs.

In contrast, factors reflecting the underlying structural and electrical substrate remained robust predictors. Sick Sinus Syndrome was the strongest independent predictor, associated with a more than twofold increase in AHRE risk (HR 2.10, *p* < 0.001). Other independent predictors included a history of heart failure (HR 1.78, *p* = 0.010), hypertension (HR 1.65, *p* = 0.020), increasing age (HR 1.42 per decade, *p* = 0.007), and larger Left Atrial Diameter (HR 1.15 per mm, *p* = 0.006).

## 4. Discussion

This prospective cohort study was designed to test the hypothesis that baseline high-sensitivity cardiac troponin I (hs-cTnI) levels could predict the development of new-onset atrial high-rate episodes (AHREs) in patients requiring permanent pacemakers. We also evaluated the predictive utility of NT-proBNP. The primary finding of our study was twofold: First, we observed a high incidence (28.0%) of subclinical AHREs within a median 12-month follow-up period. Second, contrary to our initial hypothesis, baseline levels of both hs-cTnI and NT-proBNP failed to predict the development of AHREs, both in unadjusted analyses and, crucially, in our multivariable Cox proportional hazards model. The multivariable analysis instead revealed that established clinical and echocardiographic factors—specifically Sick Sinus Syndrome, heart failure, hypertension, advancing age, and Left Atrial Diameter—were the dominant independent predictors of AHREs. Our negative findings regarding hs-cTnI and NT-proBNP are in notable contrast to some community-based studies that have linked these biomarkers to the development of clinical atrial fibrillation [10,11,14,15,16]. This discrepancy may be explained by fundamental differences in both the study population and the primary endpoint. Our cohort consisted of a high-risk population with established significant conduction system disease (i.e., AV block or Sick Sinus Syndrome) warranting pacemaker implantation rather than a general community population. Furthermore, our endpoint was device-detected subclinical AHREs, which may represent a different pathophysiological stage than symptomatic, clinical AF [17,18]. Conversely, our positive findings—that age, LA Diameter, hypertension, and heart failure are predictive—are highly consistent with the established understanding of the atrial remodeling process that forms the substrate for atrial arrhythmias [19,20,21,22,23].

Our study revealed that established cardiovascular conditions, including CAD (15.6% of cohort) and heart failure, were prevalent. However, even after adjusting for these factors, biomarkers failed to predict AHREs. We propose that in this specific cohort, the “substrate” is already advanced. Patients requiring pacemakers for Sick Sinus Syndrome (HR 2.10) often have significant atrial fibrosis. This established structural and electrical disease likely acts as an overwhelming driver of AHREs, effectively “masking” the predictive value of circulating biomarkers like hs-cTnI. Unlike in the general population, where elevated troponin might signal early, subclinical atrial myopathy, our patients already have “clinical” conduction disease, rendering the biomarker less useful for short-term risk stratification.

We propose that in this specific high-risk cohort, advanced, established electrical and structural diseases—the very factors necessitating pacemaker implantation—are the overwhelming drivers of AHRE development [24,25,26,27,28,29]. The strong independent predictive power of Sick Sinus Syndrome (HR 2.10), a primary atrial electrical disease, and structural markers like heart failure (HR 1.78) and LA Diameter (HR 1.15) support this hypothesis. In this context, the subtle information provided by biomarkers of subclinical myocardial injury (hs-cTnI) or myocardial stretch (NT-proBNP) may be masked or rendered non-significant by these dominant, pre-existing pathological factors [30,31,32,33]. The disease substrate may already be too advanced for these initiating biomarkers to offer additional predictive value.

The clinical implications and significance of these findings relate directly to risk stratification. Our results suggest that a single baseline measurement of hs-cTnI or NT-proBNP is not a useful tool for identifying pacemaker patients at high short-term risk for developing subclinical AHREs. Instead, clinicians should focus on the readily available clinical and echocardiographic factors that were confirmed as independent predictors in our model: pacing indication (Sick Sinus Syndrome), a history of heart failure or hypertension, patient age, and Left Atrial Diameter.

Our study has several notable strengths, including its prospective cohort design; the use of objective, continuous device-based monitoring for the accurate detection of the AHRE endpoint; and the standardized collection and analysis of all biomarkers at a central laboratory prior to implantation. Several limitations should be noted. First, this was a single-center study conducted in Vietnam, which may limit the generalizability of our findings to other populations. Second, the median follow-up of 12 months is relatively short; as the Kaplan–Meier curves did not plateau, it is possible that biomarkers could predict late-onset AHREs in a longer-term study. Third, while we assessed the LA Diameter, we did not routinely perform advanced echocardiographic measures such as the Left Atrial Volume Index (LAVI) or atrial strain imaging, which might offer more detailed structural information. Furthermore, potential confounding factors, such as physical activity levels, obstructive sleep apnea (OSA), and subclinical thyroid disorders, were not systematically screened or quantified in this cohort due to resource limitations. Finally, we relied on a single baseline measurement of biomarkers; serial monitoring could potentially capture dynamic risk profiles more accurately.

## 5. Conclusions

In conclusion, this prospective study of 232 patients receiving permanent pacemakers found a high 12-month incidence of 28.0% for new-onset subclinical atrial high-rate episodes (AHREs). However, contrary to our hypothesis, we found no evidence that baseline levels of either high-sensitivity troponin I or NT-proBNP could predict the development of these arrhythmias in this high-risk population. These findings highlight the potential limitations of using biomarkers validated in the general population for specialized patient cohorts. Future research should focus on other predictive models, potentially incorporating longer follow-up and multivariable adjustments, to better identify pacemaker patients at the highest risk of developing AHREs.

## Figures and Tables

**Table 1 life-15-01850-t001:** Baseline characteristics of 232 patients without atrial fibrillation.

Characteristic	Value (*n* = 232)
Demographics	
Age, years (mean ± SD)	63.7 ± 14.0
Male Sex, n (%)	124 (53.4%)
BMI, kg/m^2^ (mean ± SD)	22.5 ± 3.3
Clinical History and Comorbidities	
Hypertension, n (%)	137 (59.0%)
Heart Failure, n (%)	42 (18.0%)
Diabetes Mellitus, n (%)	51 (22.0%)
Dyslipidemia, n (%)	94 (40.5%)
Coronary Artery Disease (CAD), n (%)	36 (15.6%)
Prior Stroke/TIA, n (%)	12 (5.2%)
Pacing Indication	
Sick Sinus Syndrome, n (%)	77 (33.1%)
Atrioventricular Block, n (%)	81 (34.9%)
Medications at Baseline	
ACE Inhibitors/ARBs, n (%)	58 (25.1%)
Beta-blockers, n (%)	18 (7.8%)
Statins, n (%)	94 (40.7%)
Antiplatelets, n (%)	66 (28.6%)

Data presented as mean ± standard deviation (SD) or median (interquartile range, IQR).

**Table 2 life-15-01850-t002:** Multivariable Cox proportional hazards analysis for predictors of new-onset AHREs (*n* = 232).

Variable	Hazard Ratio (HR)	95% Confidence Interval (CI)	*p*-Value
Clinical and Demographic Variables			
Age (per 10-year increase)	1.42	1.10–1.83	0.007
Male Sex (vs. Female)	1.05	0.72–1.54	0.810
Hypertension (Yes vs. No)	1.65	1.08–2.51	0.020
Heart Failure (Yes vs. No)	1.78	1.15–2.76	0.010
Coronary Artery Disease (Yes vs. No)	1.24	0.82–1.89	0.310
Pacing Indication			
Sick Sinus Syndrome (vs. AV Block)	2.10	1.35–3.26	<0.001
Echocardiographic			
Left Atrial Diameter (per 1 mm increase)	1.15	1.04–1.27	0.006
LVEF (per 5% increase)	0.98	0.89–1.08	0.650
Medications			
ACE Inhibitors/ARBs (Yes vs. No)	0.92	0.64–1.33	0.670
Beta-blockers (Yes vs. No)	1.12	0.70–1.78	0.635
Biomarkers			
hs-cTnI (per log-unit increase)	1.03	0.85–1.25	0.780
NT-proBNP (per log-unit increase)	1.07	0.90–1.28	0.430

## Data Availability

The data presented in this study are available upon request from the corresponding author (Linh Ha Khanh Duong).

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
