# Peer review of "The Role of High-Sensitivity Troponin I in Predicting Atrial High-Rate Episodes (AHREs) in Patients with Permanent Pacemakers"

_life, 2025, doi:10.3390/life15121850_

Round 1
Reviewer 1 Report
Comments and Suggestions for Authors
In the index report, the authors explore critical and emerging area in cardiology research by investigating the role of high-sensitivity troponin I (hs-cTnI) in predicting new-onset atrial high-rate episodes (AHRE) in patients with permanent pacemakers. The study's findings suggest that in this specific cohort of patients with significant conduction system disease requiring pacemaker implantation, the advanced, established electrical and structural disease may be the overwhelming drivers of AHRE development, potentially masking the predictive value of biomarkers like hs-cTnI and NT-proBNP. These results highlight the potential limitations of using biomarkers validated in the general population for specialized patient cohorts and emphasize the need for focused risk stratification models in this high-risk population. The abstract is well written and adequality captures the salient takeaways from their research. Overall, the paper is well written and largely free of grammatical errors. While the article provides valuable insights into the potential pathophysiological mechanisms, diagnostic challenges, and treatment considerations, there are a few potential weaknesses and areas for further exploration:
- The study was conducted at a single center in Vietnam, which may limit the generalizability of the findings to other populations and healthcare settings.
- The median follow-up period was only 12 months, which is not long enough to identify predictors of late-onset AHRE. The Kaplan-Meier curve had not yet plateaued, suggesting a longer observation period could yield different results.
- The study did not account for factors like medication use, physical activity levels, or other comorbidities that could potentially influence the development of AHRE. Even CAD which is a significant predictor of AF and frequently leads to elevated trops,, at least it should have been evaluated. Adjusting for these potential confounders could have provided a more comprehensive understanding of AHRE risk.
- The study did not explore the underlying mechanisms by which the identified risk factors (e.g., sick sinus syndrome, heart failure, hypertension) may contribute to the development of AHRE. A more in-depth investigation into the pathophysiological pathways could have provided valuable insights.
- The study relied on a single baseline measurement of hs-cTnI and NT-proBNP, which may not capture the dynamic changes in these biomarkers over time. Serial measurements or a time-varying analysis could have provided a more nuanced understanding of their predictive value.
- The analysis could have included other conditions that may contribute to atrial remodeling and arrhythmia risk, such as obstructive sleep apnea, thyroid disorders, or chronic kidney disease
- While the study did assess left atrial diameter and left ventricular ejection fraction, additional echocardiographic measures of atrial and ventricular function, such as left atrial volume index, left ventricular diastolic function, or strain imaging, could have provided more comprehensive insights into the structural and functional factors associated with AHRE.
Author Response
Comment 1: “The study was conducted at a single center in Vietnam, which may limit the generalizability of the findings to other populations and healthcare settings.”
Response: We fully agree with this limitation. We have added a specific statement in the Limitations section acknowledging that as a single-center study, our findings may reflect the specific characteristics of the Vietnamese patient population and may need validation in multi-center studies involving diverse ethnic groups.
Comment 2: “The median follow-up period was only 12 months, which is not long enough to identify predictors of late-onset AHRE. The Kaplan-Meier curve had not yet plateaued, suggesting a longer observation period could yield different results.”
•
Response: We acknowledge that a 12-month follow-up is relatively short for observing late-onset arrhythmias. We have revised the Discussion to clarify that our findings pertain primarily to short-term risk stratification. We have also emphasized in the Conclusion that biomarkers might still hold value for long-term prediction, suggesting that future research should incorporate longer observation periods as the event rate continues to rise.
Comment 3: “The study did not account for factors like medication use, physical activity levels, or other comorbidities that could potentially influence the development of AHRE. Even CAD which is a significant predictor of AF and frequently leads to elevated trops, at least it should have been evaluated. Adjusting for these potential confounders could have provided a more comprehensive understanding of AHRE risk.”
•Response: We appreciate this critical feedback.
â–ªComorbidities & Medications: We have re-analyzed our raw data and updated Table 1 to include the prevalence of Coronary Artery Disease (CAD) (15.6%), as well as the use of Beta-blockers (7.8%), ACEi/ARBs (25.1%), and Statins (40.7%).
â–ªMultivariable Analysis: We updated Table 2 (Multivariable Cox Regression) to include CAD and medications as covariates. The analysis showed that these factors were not significant predictors, reinforcing our conclusion that the structural substrate (Sick Sinus Syndrome, LA diameter) is the dominant driver of AHRE in this cohort.
â–ªPhysical Activity: We acknowledge that physical activity levels were not systematically quantified in this study. We have noted this as a limitation.
Comment 4: “The study did not explore the underlying mechanisms by which the identified risk factors (e.g., sick sinus syndrome, heart failure, hypertension) may contribute to the development of AHRE. A more in-depth investigation into the pathophysiological pathways could have provided valuable insights”
•Response: We have expanded the Discussion section to elaborate on the pathophysiology. We discuss how Sick Sinus Syndrome and Heart Failure represent an advanced stage of “atrial cardiomyopathy,” characterized by fibrosis and structural remodeling. We propose that this established substrate “masks” the subtle signals of myocardial injury that hs-cTnI typically detects in healthier populations.
Comment 5: “The study relied on a single baseline measurement of hs-cTnI and NT-proBNP, which may not capture the dynamic changes in these biomarkers over time. Serial measurements or a time-varying analysis could have provided a more nuanced understanding of their predictive value.”
•Response: We agree that serial measurements would provide better insight into the dynamic risk profile. We have added this point to the Limitations section, suggesting that future studies should investigate dynamic changes in biomarker levels (delta-troponin) rather than a single snapshot.
Comment 6: “The analysis could have included other conditions that may contribute to atrial remodeling and arrhythmia risk, such as obstructive sleep apnea, thyroid disorders, or chronic kidney disease”
•Response:
â–ªCKD: Patients with severe renal impairment (eGFR ≤ 30 mL/min/1.73m²) were explicitly excluded from our study to avoid confounding effects on troponin levels (Methods section).
â–ªThyroid/OSA: We did not systematically screen for OSA or subclinical thyroid disorders in this cohort due to resource limitations. We have acknowledged this in the Limitations section.
Comment 7: “While the study did assess left atrial diameter and left ventricular ejection fraction, additional echocardiographic measures of atrial and ventricular function, such as left atrial volume index, left ventricular diastolic function, or strain imaging, could have provided more comprehensive insights into the structural and functional factors associated with AHRE.”
•Response: We appreciate this suggestion. Our study relied on standard echocardiographic parameters (LA diameter, LVEF) that are routinely collected in clinical practice in Vietnam. While we did not perform strain imaging or calculate LAVI for all patients, we have listed this as a limitation and a recommendation for future research to provide more granular structural insights
Reviewer 2 Report
Comments and Suggestions for Authors
The manuscript addresses an important clinical question and is clearly written, with a well-defined methodology and coherent results. The finding—lack of predictive value of high sensitive cardiac troponin I(hs-cTnI) for atrial high-rate episodes (AHRE) in pacemaker patients—is clinically relevant. However, several methodological, interpretative, and structural issues need to be addressed to strengthen the paper.
Major Points
#1
The manuscript alternates between describing 272 and 232 patients. While the at-risk cohort is correctly defined, the flow of exclusions and final analytic population should be clearly represented (e.g., a CONSORT-style diagram). This improves transparency and allows readers to assess selection bias.
#2
Although AHRE definitions from AHA guidelines are cited, the manuscript lacks specifics on:
- minimum episode duration required for inclusion,
- whether all intracardiac electrograms were reviewed by two independent observers,
- inter-observer agreement.
These factors strongly influence AHRE incidence and the validity of the primary endpoint.
#3
The discussion suggests that biomarkers may be “masked” by advanced structural disease, but offers no sensitivity analyses (e.g., interaction terms, stratification by pacing indication, renal function, or left atrium size). Without such analyses, this explanation remains speculative. Strengthening the argument with additional analyses or explicitly acknowledging the uncertainty is recommended.
#4
The manuscript acknowledges that AHRE incidence had not plateaued by 12 months (Figure 1). However, this has major implications: the biomarkers tested may require longer observation to manifest predictive value. This limitation deserves a deeper and more prominent discussion, as it directly affects the central conclusion.
Minor Points
#1
The Kaplan-Meier curves (Figures 1 and 2) lack clear axis labels, confidence intervals, and number-at-risk tables. These should be added to meet typical journal standards.
#2
The abstract repeats some numerical values unnecessarily and should explicitly state the key negative result earlier, improving readability and impact.
Thank you for the opportunity to review this manuscript. It addresses a clinically meaningful question, and I appreciate the authors’ efforts in conducting a prospective study in a challenging patient population.
Comments on the Quality of English LanguagePlease see my comments to the authores as above.
Author Response
Major Point 1: “The manuscript alternates between describing 272 and 232 patients. While the at-risk cohort is correctly defined, the flow of exclusions and final analytic population should be clearly
represented (e.g., a CONSORT-style diagram). This improves transparency and allows readers to assess selection bias.”
•Response: We apologize for the confusion. The total eligible cohort was 272 patients. From this group, 40 patients with documented pre-existing AF were excluded from the prospective analysis to form the final “at-risk” cohort of 232 patients. We have revised the Methods (Study Design) and Results sections to clearly delineate this flow, ensuring transparency regarding the analytic population.
Major Point 2: “Although AHRE definitions from AHA guidelines are cited, the manuscript lacks specifics on:
•minimum episode duration required for inclusion,
•whether all intracardiac electrograms were reviewed by two independent observers,
•inter-observer agreement.
These factors strongly influence AHRE incidence and the validity of the primary endpoint.”
Response: We have updated the Methods section to specify that:
1. AHRE was defined as device-detected atrial tachyarrhythmias with a rate >175 bpm.
2. To ensure validity, all stored intracardiac electrograms (EGMs) were reviewed by two independent cardiologists to rule out artifacts or far-field R-wave sensing.
3. Disagreements were resolved by consensus or consultation with a third senior electrophysiologist.
Major Point 3: “The discussion suggests that biomarkers may be “masked” by advanced structural disease, but offers no sensitivity analyses (e.g., interaction terms, stratification by pacing indication, renal function, or left atrium size). Without such analyses, this explanation remains speculative. Strengthening the argument with additional analyses or explicitly acknowledging the uncertainty is recommended.”
•Response: To address this, we performed an expanded multivariable analysis (Revised Table 2) incorporating potential confounders such as Coronary Artery Disease and medication use. The results remained consistent: biomarkers were not predictive, while structural/electrical factors (Sick Sinus Syndrome, Heart Failure) were. This supports
our “masking” hypothesis - that the advanced disease substrate outweighs other factors. We have updated the Discussion to reflect these robust findings.
Major Point 4: “The manuscript acknowledges that AHRE incidence had not plateaued by 12 months (Figure 1). However, this has major implications: the biomarkers tested may require longer observation to manifest predictive value. This limitation deserves a deeper and more prominent discussion, as it directly affects the central conclusion.”
•Response: We have added a paragraph to the Discussion addressing the time-dependent nature of arrhythmia development. We explicitly state that our negative findings apply to short-term risk (1 year) and that biomarkers might still be relevant for predicting late-onset remodeling and arrhythmia in a longer follow-up setting.
Minor Point 1: “The Kaplan-Meier curves (Figures 1 and 2) lack clear axis labels, confidence intervals, and number-at-risk tables. These should be added to meet typical journal standards.”
•Response: We have utilized the MDPI Figure Editing service to ensure that the figures comply with the journal’s guidelines.
Minor Point 2: “The abstract repeats some numerical values unnecessarily and should explicitly state the key negative result earlier, improving readability and impact.”
•Response: We have rewritten the Abstract to be more concise. We removed redundant numerical repetitions and moved the primary negative finding (lack of predictive value of hs-cTnI) to the beginning of the Results section in the Abstract to improve clarity and immediate impact.
Round 2
Reviewer 1 Report
Comments and Suggestions for Authors
The authors have satisfactorily responded to most concerns raised during the previous review. I have no further suggestions.
Reviewer 2 Report
Comments and Suggestions for Authors
Thank you for the opportunity to review the revised manuscript. The authors have addressed all of my previous questions and concerns appropriately. I have no further comments.
Comments on the Quality of English LanguagePlease see my comments to the authores as above.